# Actor–Partner Effects of Personality Traits and Psychological Flexibility on Psychological Distress Among Couples Coping with Cancer

**DOI:** 10.3390/bs14121161

**Published:** 2024-12-04

**Authors:** Leegal Bar-Moshe-Lavi, Nimrod Hertz-Palmor, Keren Sella-Shalom, Michal Braun, Noam Pizem, Einat Shacham-Shmueli, Eshkol Rafaeli, Ilanit Hasson-Ohayon

**Affiliations:** 1Department of Psychology, Bar-Ilan University, Ramat Gan 5290002, Israel; sellake@biu.ac.il (K.S.-S.); eshkol@biu.ac.il (E.R.); ilanit.hasson-ohayon@biu.ac.il (I.H.-O.); 2MRC Cognition and Brain Sciences Unit, University of Cambridge, Cambridge CB2 1TN, UK; nimrod.hertz@mrc-cbu.cam.ac.uk; 3Sackler School of Medicine, Tel Aviv University, Tel Aviv-Yafo 6997801, Israel; bmichal@mta.ac.il (M.B.); einat.shachamshmueli@sheba.health.gov.il (E.S.-S.); 4School of Behavioral Sciences, The Academic College of Tel Aviv-Jaffo, Tel Aviv-Yafo 6997712, Israel; 5Chaim Sheba Medical Center at Tel Hashomer, Ramat Gan 5262000, Israel; noam.pizem@sheba.health.gov.il

**Keywords:** actor–partner, personality traits, psychological flexibility, cancer, depression, anxiety, psycho-oncology

## Abstract

In this study, we applied the actor–partner interdependence model (APIM) to explore the associations between personality traits (Big Five) and psychological flexibility, on the one hand, and depression and anxiety, on the other hand, among patients with cancer and their spouses. Method: Forty-six patient—spouse dyads (N = 92) completed the anxiety and depression scales from the Patient-Reported Outcomes Measurement Information System (PROMIS), the ten-item personality inventory (TIPI), and the psychological flexibility scale (AAQ-2). Multilevel APIM models, adjusted for multiple testing, showed that neuroticism and psychological flexibility had actor effects on patients’ depression and anxiety. Furthermore, neuroticism had actor effects on spouses’ depression and anxiety, and agreeableness had actor effects on spouses’ anxiety. In addition, patients’ psychological flexibility and neuroticism had partner effects on spouses’ depression. Conclusion: Being psychologically flexible but emotionally stable is important for one’s own and one’s partner’s psychological outcomes in the context of dyadic coping with cancer. Implications include informing couples’ therapists in the context of psycho-oncology on the importance of considering personality traits and improving psychological flexibility.

## 1. Introduction 

### 1.1. Psychological Outcomes Among Couples Coping with Cancer

Cancer patients and their spouses are at high risk of developing psychological distress at different stages of coping with the illness [1,2]. Approximately 40% of patients and spouses report high levels of anxiety, depression, general psychological distress, and low quality of life near the time of diagnosis [3]. At later stages during treatment, both cope with diverse cancer-related challenges, such as the intensive treatments patients receive and the emotional and instrumental support that spouses are often required to provide [4]. Psychological distress, post-diagnosis, is estimated to be relatively high among patients, and the most prevalent forms of distress are major depression (15%), minor depression (20%), and anxiety (10%) [5]. Further, the data show a significant incidence of emotional distress during survivorship (up to three years after diagnosis) for the recovering individual (depression 11.6%, anxiety 17.9%) and their spouse (depression 10.6%, anxiety 13.9%) [6]. The fact that both partners exhibit high levels of psychological distress, from diagnosis to survivorship, reinforces the need for a comprehensive understanding of the psychological consequences of cancer in a dyadic context [7].

Meta-analyses and systematic reviews have highlighted patient–spouse partner effects when considering psychological outcomes. For example, Hagedoorn et al. (2008) [8] showed a moderate association between patient and partner distress in their meta-analysis. In addition, Traa et al. (2015) [9], in their systematic review, showed that, in the context of coping together as a couple, synchronization in coping style and method of coping were related to positive aspects of marital functioning, such as open communication. Additionally, a literature review by Chen et al. (2021) [10] showed that negative dyadic coping (i.e., providing ambivalent support, hostile communication, and hiding information concerning cancer) was associated with a greater number of depression symptoms and poor quality of life for both partners. A recent systematic review by Hasson-Ohayon et al. (2022) [11] showed that patterns of mutual communication regarding cancer-related issues were related to better outcomes for both partners. 

Although the dyadic effects of couples coping with cancer have been extensively studied using outcome and relational variables, little is known about the effects of personality traits in this context. Personality refers to emotional and behavioral tendencies that may affect how individuals cope with health challenges [12]. In the current study, we focused on personality characteristics as possible correlates of depression and anxiety when coping with cancer in a dyadic context. Specifically, we explored the association between personality traits (Big Five) and psychological flexibility, on the one hand, and depression and anxiety on the other hand, among patients and spouses, using an actor–partner interdependence model (APIM).

### 1.2. Personality Traits, Psychological Flexibility, and Psychological Distress

Personality traits represent consistent thinking patterns and actions over time. They refer to persistence and personal motivation regarding an individual’s feelings, attitudes, and behavior [13]. A popular conceptualization of personality traits includes five main traits (the Big Five): Openness (to experience), Conscientiousness, Extraversion, Agreeableness, and Neuroticism (Emotional Stability) [14,15].

According to the Big Five model, the five traits represent personality at the broadest level, each containing more specific personality characteristics. Neuroticism represents the tendency to have a temperament affected by negative emotions such as sadness, anxiety, anger, and stress. Conversely, a person who does not have a neurotic temperament has a tendency to be highly emotionally stable. Extraversion includes qualities such as sociability, assertiveness, and positive sensitivity. Conscientiousness, for its part, is related to a reliable personality and includes social impulse control (e.g., delaying gratification, following norms and rules, and planning and organizing tasks). Agreeableness reflects the tendency and motivation to have good relations with others and includes sympathy, trust, gentleness, and altruism. Finally, openness to experience combines cognitive flexibility, sensitivity to aesthetics, depth, breadth, and originality of mental experience. This trait relates to the continuous search for new experiences, ideas, and creativity [14,15].

Personality is associated with physical health and health behaviors. For example, higher conscientiousness and lower neuroticism are associated with better physical health (measured by disease severity) and health behaviors (i.e., more exercise, healthier diet, less substance use) among general population and cancer patients [16]. In addition, personality traits are related to individuals’ quality of life and ability to cope with change and crisis. A systematic review in which this claim was examined revealed that personality traits were correlated with patients’ quality of life in the context of various illnesses (including cancer) [17]. For example, a study in which the effects of personality traits on cancer patients’ quality of life were examined showed that high conscientiousness and extraversion, and low neuroticism, predicted a better quality of life [12]. Personality traits have also been found to be related to the coping patterns of individuals with cancer. For example, whereas extraversion was shown to be positively correlated with adaptive coping with cancer, neuroticism was found to be positively associated with avoidant coping [18].

Little research has been done on the dyadic effect of personality traits in the context of coping with cancer. To the best of our knowledge, only two studies in which an actor–partner approach was applied examined personality traits in this context. The findings of Hamidou et al. (2018) [19] showed a partner effect for conscientiousness. Namely, the patient’s conscientiousness was negatively related to the quality of life of the patient’s partner and vice versa. Second, a study by Wang et al. (2022) [2] focused on personality traits, acceptance of the illness, and the association between these factors and depression. The patient’s neuroticism was found to have a positive association with the spouse’s depression, and the patient’s acceptance of the illness mediated this association. In contrast, each partner’s extraversion was associated with their own greater illness acceptance, which further reduced the level of depression for each partner. Moreover, the extraversion of the spouses was found to be negatively related to the patients’ depression level and a mediator of the spouses’ illness acceptance. On the basis of these two studies, it seems that being more emotionally stable, more social, and more adherent to norms may be beneficial for both patients and spouses in the context of coping with cancer.

In addition to the possible role of personality traits in coping with cancer, psychological flexibility is also an important construct to consider in this context. Psychological flexibility represents the individual’s ability to be present in the moment, with their feelings and thoughts, without defenses, and in accordance with reality [20]. This quality includes the ability to make behavioral changes that allow the pursuit of the individual’s values and goals. In contrast, psychological inflexibility includes rigidity in responses, a reduction in being present at the moment, and a reduction in the likelihood of taking value-based actions [21] Openness to experience, one of the Big Five personality traits, involves some aspect of cognitive flexibility, like the willingness to engage with new ideas. However, the main characteristics of this trait include having an imagination, curiosity, and a preference for variety [14]. Psychological flexibility, on the other hand, refers to the ability to adapt to situational demands and maintain balance in the face of challenges via the acceptance of thoughts and feelings without judgment and aligning actions with personal values [21].

Studies have shown that increased flexibility among patients with cancer is significantly related to reduced anxiety and depression [22]. Indeed, the use of psychological flexibility training has been associated with the ability to adapt to the cancer diagnosis and to develop more self-awareness, thus reducing psychological distress [23]. Nevertheless, despite the importance of psychological flexibility for coping with cancer and the understanding that dyadic effects are significant predictors of each partner’s personal coping ability, there have been no studies (to the best of our knowledge) in which the dyadic effects of psychological flexibility in this context have been examined.

As mentioned above, it has been well-established that a cancer diagnosis affects both patients and spouses (e.g., [2]). Therefore, dyadic approaches, such as the APIM, in which both partners’ effects on the other are assessed, seem most suitable for examining the psychological effects of cancer. In the current study, we used the APIM in order to explore the associations between patients’ and spouses’ personality traits and psychological flexibility and their own and their partners’ depression and anxiety.

## 2. Method

### 2.1. Study Design and Participants

The current study was part of a large-scale research study in which dyadic communication behaviors in psycho-oncology were examined. Inclusion criteria were (1) patients coping with various cancer types and their spouses; (2) participants over 18 years of age; (3) couples in a committed relationship; (4) no comorbidity with severe cognitive or mental disorders or severe organic diseases; and (5) a proper understanding of the Hebrew language. Following the Sened et al. (2020) [24] study, which investigated dyads using the actor–partner interdependence model (APIM) analysis in the context of physical illnesses with a sample of 42 participants, we determined a sample size of at least 40 dyads. Participants in the current study were 46 heterosexual patient–spouse dyads (92 individuals) who took part in the cross-sectional stage of the project. They completed the anxiety and depression scales from the Patient-Reported Outcomes Measurement Information System (PROMIS), the ten-item personality inventory (TIPI), and the Acceptance and Action Questionnaire (AAQ-2) for the assessment of psychological flexibility. In order to try to avoid bias in the study, we also contacted all the patients whom the staff approved in this regard during the recruitment period.

### 2.2. Procedure

The recruitment procedures and study protocol received the approval of the institutional review board (IRB) of Sheba Medical Center (approval no 7673-20-SMC). The Israel Cancer Association funded the study, and participants were remunerated for their participation (50 US dollars $ per couple). The recruitment process included contacting potential participants during their stays in the hospital and having them submit anonymous online surveys after they had provided consent.

### 2.3. Measurements

Ten-item personality inventory (TIPI). This measure is a 10-item measure of the Big Five personality trait dimensions [25]. Responses are coded on a 7-point Likert scale, ranging from “don’t agree at all” to “largely agree”, with higher scores reflecting a high tendency to exhibit the personality trait at hand.

The Acceptance and Action Questionnaire (AAQ-2). This 7-item self-report was used to measure psychological flexibility [21]. Responses are coded on a 7-point Likert scale, ranging from “never true” to “always true”, with lower scores expressing more psychological flexibility. A sample item is: “Emotions cause problems in my life”. The Cronbach’s alpha coefficient in our data was high, α = 0.81. The scale was reversed in the statistical analysis so that higher scores represented greater flexibility.

Patient-Reported Outcomes Measurement Information System (PROMIS). The scale consists of 16 self-reported items, with 8 items for each domain (depression/anxiety). Responses are coded on a 5-point Likert scale, ranging from “never” to “all the time”. A sample item from the anxiety domain is, “I found it hard to focus on anything other than my anxiety”, and for the depression domain, “I felt that I had nothing to look forward to”. The PROMIS scale was developed collaboratively between the National Institutes of Health (NIH) and academic researchers and is viewed as a psychometrically acceptable tool. We used the Hebrew translation of this scale [26]. In line with scoring manuals, scores for each PROMIS measure are normalized to a mean of 50 and a standard deviation of 10, and all PROMIS raw scores are converted to T-scores. In the present study, the Cronbach’s alpha coefficients were high (α = 0.91 for anxiety, α = 0.88 for depression).

### 2.4. Statistical Analysis

We used the APIM to test actor and partner effects among patients and spouses. In accordance with recommendations regarding APIM best practice [27]. APIM can be modelled via multiple statistical techniques, and most recommended are Structural Equation Modelling (SEM), and Multilevel Modelling, also known as Mixed-Effects Modelling. Mixed-effects models account for both fixed effects (which apply to the entire population) and random effects (which account for variability within groups or individuals) in hierarchical or clustered data, such as the couples in our sample [28]. We constructed our model as a multilevel model where fixed effects were the interaction between participant’s role (patient or spouse) and the variables of interest (Big Five personality traits and psychological flexibility), and the random effect was number of couples. Each variable of interest produced four effects: patient’s actor effect (patients on themselves), patient’s partner effect (spouses on patients), spouse’s actor effect (spouses on themselves), and spouse’s partner effect (patients on spouses). We conducted this analysis using the lmerTest package in R [29]. As some of our variables of interest were intercorrelated, we performed separate models to overcome multicollinearity, so that in each model a different Big-5 factor was introduced as an independent fixed-effect, and either anxiety or depression was introduced as the fixed effect. A second justification for performing separate models was that introducing all factors into one model would lead to a non-optimal observations-to-variables ratio. We controlled for multiple comparisons with the false discovery rate (FDR), both within each model and across all models [30], using the stats package in R (version 4.3.0) [31]. Results were considered significant for α < 0.05, after FDR adjustment.

## 3. Results

### 3.1. Sample Description

Of the 46 couples who participated in the study, 28 (60%) couples included a male patient and a female spouse, whereas 18 (40%) couples were the opposite. The mean age was 61.07 for patients (SD = 12.22) and 59.39 for spouses (SD = 11.59). The mean length of the relationship was 30.65 years (SD = 15.45). Of all the participants (N = 92), 39 participants (42.4%) reported that they had a college or university education, and 28 (30.4%) either had an elementary school, high school, or diploma education. Among the patients, half had been diagnosed with gastrointestinal cancer (52.2%), whereas the rest were diagnosed with other types (47.8%). Moreover, 39.1% of the patients reported stage IV illness, 19.6% reported stage III, and the others reported stage I, II, or unknown. The mean time since diagnosis was 5 years (SD = 2.21).

For clarity, psychological flexibility scores, represented by the AAQ-2, were inverse, so that higher scores represent greater flexibility.

Table 1 and Table 2 show the full results of the APIMs.

### 3.2. Patients’ Depression

Actor effects. APIM models showed significant actor effects of patients’ neuroticism (B = 0.49, 95% confidence interval (CI) = 0.24,0.74, p_adjusted_ = 0.003) and psychological flexibility (B = −0.52, 95% CI = −0.78, −0.26, p_adjusted_ = 0.003) on patients’ depression (Figure 1).

Partner effects. Partners’ personality traits and psychological flexibility had no significant effect on patients’ depression (Figure 1).

### 3.3. Patients’ Anxiety

Actor effects. Patients’ anxiety was significantly linked to neuroticism (B = 0.70, 95% CI = 0.48, 0.93, p_adjusted_ < 0.001) and psychological flexibility (B = −0.56, 95% CI = −0.85, −0.28, p_adjusted_ = 0.003). Openness to experience was initially negatively associated with patients’ anxiety (B = −0.37, 95% CI = −0.67, −0.07, p_unadjusted_ = 0.021), but became insignificant after FDR adjustment (Figure 2).

Partner effects. Akin to the findings regarding patients’ depression, partners’ traits had no significant effect on patients’ anxiety (Figure 2).

### 3.4. Spouses’ Depression

Actor effects. Significant actor effects were observed for spouses’ neuroticism (B = 0.38, 95% CI = 0.12,0.64, p_adjusted_ = 0.026) and agreeableness (B = −0.49, 95% CI = −0.78, −0.19, p_adjusted_ = 0.012) on spouses’ depression (Figure 3).

Partner effects. Significant effects on spouses’ depression were observed for their partners’ (i.e., the patients’) neuroticism (B = 0.37, 95% CI = 0.13, 0.62, p_adjusted_ = 0.022) and psychological flexibility (B = −0.44, 95% CI = −0.70, −0.18, p_adjusted_ = 0.012) (Figure 3).

### 3.5. Spouses’ Anxiety

Actor effects. Neuroticism was the only actor effect significantly associated with spouses’ anxiety (B = 0.39, 95% CI = 0.15, 0.62, p_adjusted_ = 0.019) (Figure 4).

Partner effects. Partners’ neuroticism was initially significantly associated with spouses’ anxiety (B = 0.27, 95% CI = 0.05,0.50, p_unadjusted_ = 0.021), but was insignificant following FDR adjustment (Figure 4).

## 4. Discussion

In this study, we aimed to expand the existing knowledge regarding the involvement of personality characteristics in dyadic coping with cancer. To this end, we examined the associations between patients’ and spouses’ personality traits and psychological flexibility and their own and their spouses’ depression and anxiety.

Results showed that the actor effects of patients’ neuroticism and psychological flexibility were associated with their own depression and anxiety outcomes, and their openness to experience was negatively and initially associated, before adjustment for FDR, with their own anxiety. That is, the less neurotic (i.e., more stable) and more psychologically flexible the patient, the less depression and anxiety they reported; furthermore, the more open to experience they were, the less anxiety they (the patient) reported. In addition, the actor effect of spouses’ neuroticism was positively associated with their own depression and anxiety, and their agreeableness was negatively associated with their own depression. That is, the less neurotic and more agreeable the spouse, the less depression and anxiety they reported. No actor effect was found for spouses’ psychological flexibility on depression and anxiety outcomes. Furthermore, no significant partner effect was found for either personality traits or psychological flexibility on patients’ depression and anxiety. However, for the spouses’ partner effect, results showed an association between, on the one hand, patients’ psychological flexibility and neuroticism and, on the other hand, spouses’ depression. In addition, patients’ neuroticism had a marginal association, after adjustment for FDR, with spouses’ anxiety. Hence, psychological flexibility was associated positively, and neuroticism was associated negatively with spouses’ depression and anxiety.

The results regarding the associations between neuroticism and psychological outcomes are in line with findings from previous studies, which showed that patients and spouses with low levels of neuroticism expressed less depression and anxiety when coping with cancer (e.g., [32,33]). This finding has also been found among the general population [34]. Namely, people who are less neurotic and more emotionally stable seem less inclined to experience negative emotions, such as sadness, anxiety, and stress [35]. There is, therefore, a lower expectation that they will experience high levels of distress, even when coping with a life-threatening illness such as cancer [36]. Likewise, the finding that openness to experience was negatively related to patient’s anxiety is also in line with past research [37]. Although this finding was not sustained in the current study after the FDR modifications, it points to the possible role of openness to experience in patients’ psychological well-being. For example, openness to experience has been found to be associated with more participation in alternative treatments (e.g., yoga, meditation, homeopathy), and this participation has been found to contribute to psychological well-being among cancer patients [38]. Furthermore, our finding that psychological flexibility was associated with psychological outcomes is also in line with findings from previous studies—that is, patients with high psychological flexibility have reported less depression and anxiety when coping with cancer (e.g., [39,40]). People who are flexible tend to be more aware and less rigid, and such flexibility may assist them in adapting to crisis situations [23].

Moreover, the finding that spouses’ agreeableness was negatively associated with depression is supported by a previous study in which caregivers of sick elderly people were examined, and agreeableness was found to be associated with less psychological distress and less loneliness [41]. Highly agreeable people tend to be altruistic and to create social relationships characterized by trust and empathy, helping them to develop a good support system, which is important for coping with cancer [42]. These findings point to the important role played by personality traits in coping with cancer.

Contrary to previous studies, whose findings indicated that patients tended to be more affected by their partners than vice-versa (e.g., [41,43]), no partner effect was found in the current study for patients. This lack of finding can perhaps be attributed to the characteristics of the current study’s sample—namely, 60% of the sample comprised female partners of male patients, and there was diversity among the diagnoses, both in terms of cancer stage and cancer type. Gender and role may affect the partners’ effect [3], and the heterogenicity of the sample may also affect the partners’ effect. For example, Linden et al. (2012) [44] found differences in intensities of psychological distress according to cancer type. In the current study, we observed significant partner effects of patients’ psychological flexibility and neuroticism on spouses’ depression. This finding suggests that patients’ emotional stability and flexibility when confronting challenges is important for their spouses’ experience. The spouse, as a caregiver, might feel that the task of reducing the patient’s stress falls solely on them [45]. When this task is decreased (in the case of a less neurotic and more flexible patient), the spouse may experience less burden.

## 5. Limitations

The study’s limitations include both the recruitment method and the number of participants. First, many potential participants (i.e., who were approached during their hospital visits/stays) had difficulty cooperating, given their health conditions. Therefore, there may be a bias in the study toward those who agreed to participate; those who agreed to participate may be characterized by more emotional stability and psychological flexibility than those who did not. Second, as the study was cross-sectional, it is not possible to draw conclusions on causality. The heterogeneity of cancer type and stage may also have affected patients’ and spouses’ psychological flexibility; that is, this characteristic is a variable that can change throughout a person’s life. Our sample included 92 participants, which was sufficient in terms of statistical powers, as evidenced by the significant effects achieved in our analyses, even following *p*-value adjustments for multiple comparisons. However, these results should also be viewed in terms of external validity, i.e., our ability to generalize conclusions to a broader population. The application of our results to couples struggling with cancer should be interpreted with caution, due to the relatively small sample size. Another implication of our small sample size is that higher-order interactions, such as the potential moderation of actor and partner’s psychological flexibility on the association between neuroticism and mental health outcomes, could not be investigated with sufficient statistical power. Therefore, it is reasonable to suggest that, in a larger sample, and in an analysis in which the diversity of the groups is taken into consideration, it would be possible to obtain additional significant results and examine complex relationships between multiple variables and their dynamics.

## 6. Summary and Conclusions of the Study

Our results indicate that personality characteristics are related to emotional distress among patients and spouses coping with cancer. These findings strengthen the idea that partners are influenced by each other and emphasize the importance of knowledge and interventions related to personality characteristics in dyadic coping with cancer. Specifically, personality characteristics should be taken into consideration during psychological screening and interventions. For example, identifying neuroticism and inflexibility at early stages of the disease, even as early as the cancer diagnosis, can assist in tailoring a therapeutic approach that emphasizes self-acceptance and adaptive coping, such as acceptance and commitment therapy (ACT), which is a beneficial approach in psycho-oncology to increase psychological flexibility [46]. Considering psychological flexibility from a dyadic perspective might be particularly important in couples therapy, given the partner effect. Accordingly, promoting joint flexible coping might be beneficial for couples coping with cancer. Our findings align with models emphasizing the importance of personality traits in understanding psychological outcomes in medical contexts. For instance, the “Interactional stress moderation models” highlight the influence of personality on both exposure to stressful life circumstances and the availability of coping resources [28].

## Figures and Tables

**Figure 1 behavsci-14-01161-f001:**
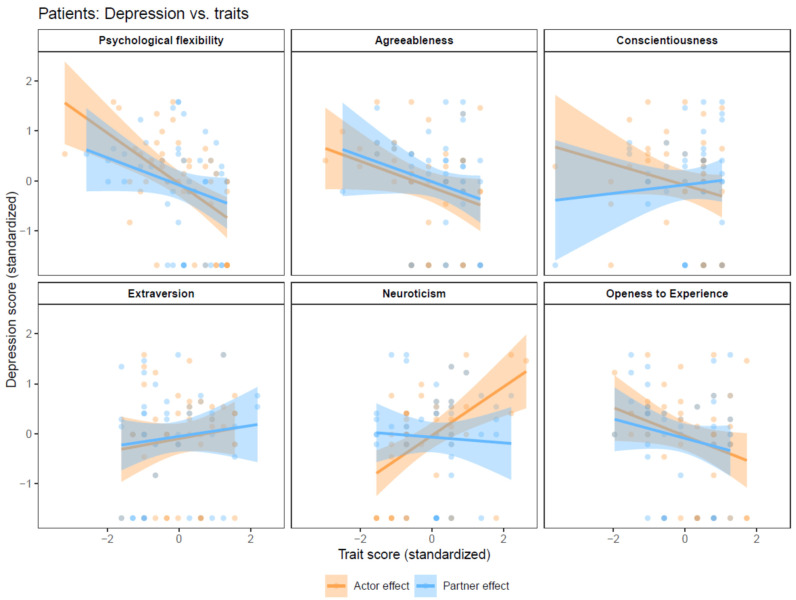
Actor–partner interdependence models with depression as the dependent variable.

**Figure 2 behavsci-14-01161-f002:**
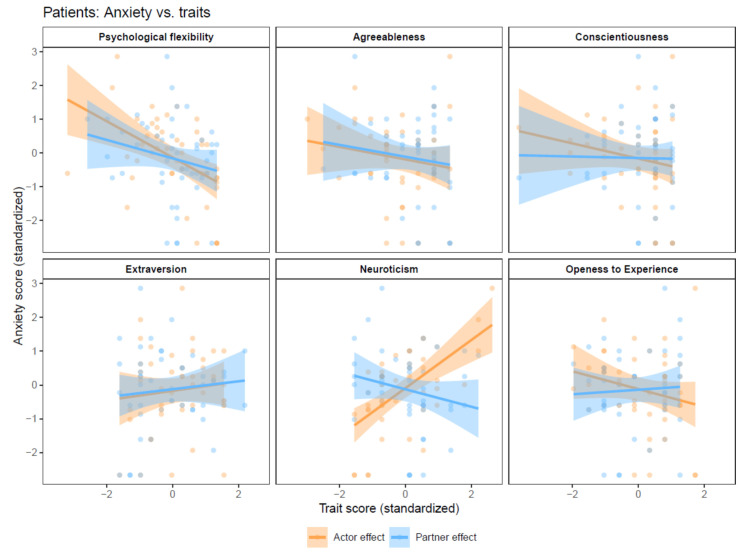
Actor-Partner Interdependence models with anxiety as the dependent variable.

**Figure 3 behavsci-14-01161-f003:**
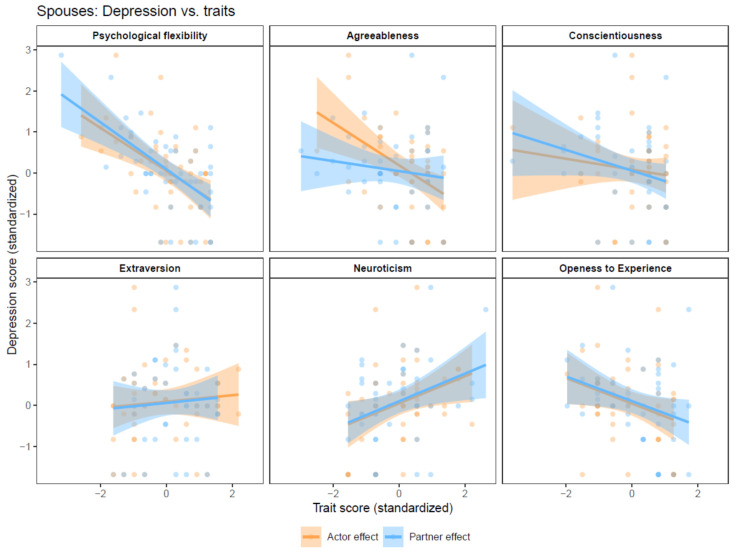
Actor-Partner Interdependence models with depression as the dependent variable.

**Figure 4 behavsci-14-01161-f004:**
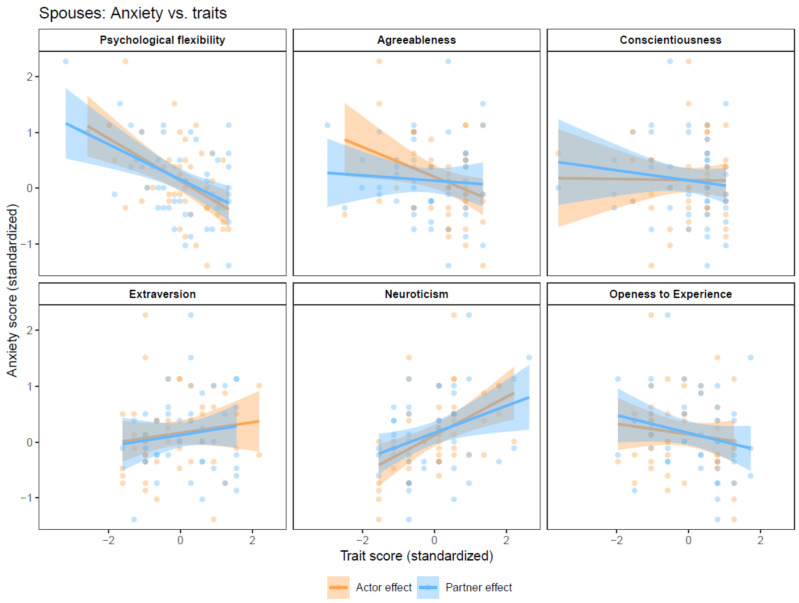
Actor-Partner Interdependence models with anxiety as the dependent variable.

**Table 1 behavsci-14-01161-t001:** Actor–partner Interdependence models with depression as the dependent variable.

Independent Variable	APIM Effect	Dependent Variable	B (95% CI)	P_Unadjusted_	P_Adjusted Across Model_	P_Adjusted Across All_
**Neuroticism**						
Patient’s neuroticism	Actor	Patient’s depression	0.49 (0.24,0.74)	0.000	0.001	0.003
Spouse’s neuroticism	Actor	Spouse’s depression	0.38 (0.12,0.64)	0.006	0.011	0.026
Spouse’s neuroticism	Partner	Patient’s depression	0.00 (−0.26,0.26)	0.99	0.99	0.99
Patient’s neuroticism	Partner	Spouse’s depression	0.37 (0.13,0.62)	0.005	0.011	0.022
**Extraversion**						
Patient’s extraversion	Actor	Patient’s depression	0.08 (−0.24,0.41)	0.63	0.97	0.73
Spouse’s extraversion	Actor	Spouse’s depression	0.04 (−0.25,0.34)	0.78	0.97	0.85
Spouse’s extraversion	Partner	Patient’s depression	0.10 (−0.20,0.39)	0.53	0.97	0.73
Patient’s extraversion	Partner	Spouse’s depression	0.08 (−0.24,0.41)	0.63	0.97	0.73
**Agreeableness**						
Patient’s agreeableness	Actor	Patient’s depression	−0.24 (−0.49,0.01)	0.07	0.12	0.18
Spouse’s agreeableness	Actor	Spouse’s depression	−0.49 (−0.78,−0.19)	0.002	0.010	0.012
Spouse’s agreeableness	Partner	Patient’s depression	−0.29 (−0.58,0.01)	0.06	0.12	0.18
Patient’s agreeableness	Partner	Spouse’s depression	−0.14 (−0.39,0.11)	0.29	0.36	0.46
**Conscientiousness**						
Patient’s conscientiousness	Actor	Patient’s depression	−0.21 (−0.48,0.06)	0.14	0.34	0.27
Spouse’s conscientiousness	Actor	Spouse’s depression	−0.11 (−0.41,0.19)	0.48	0.69	0.72
Spouse’s conscientiousness	Partner	Patient’s depression	0.09 (−0.20,0.39)	0.55	0.69	0.73
Patient’s conscientiousness	Partner	Spouse’s depression	−0.25 (−0.52,0.02)	0.08	0.34	0.18
**Openness to experiences**						
Patient’s openness to experiences	Actor	Patient’s depression	−0.26 (−0.56,0.04)	0.10	0.33	0.21
Spouse’s openness to experiences	Actor	Spouse’s depression	−0.23 (−0.54,0.08)	0.15	0.33	0.28
Spouse’s openness to experiences	Partner	Patient’s depression	−0.08 (−0.38,0.23)	0.64	0.80	0.73
Patient’s openness to experiences	Partner	Spouse’s depression	−0.20 (−0.50,0.10)	0.20	0.33	0.34
**Psychological flexibility**						
Patient’s psychological flexibility	Actor	Patient’s depression	−0.52 (−0.78,−0.26)	<0.001	0.001	0.003
Spouse’s psychological flexibility	Actor	Spouse’s depression	−0.28 (−0.56,0.01)	0.07	0.12	0.18
Spouse’s psychological flexibility	Partner	Patient’s depression	0.02 (−0.27,0.31)	0.88	0.89	0.92
Patient’s psychological flexibility	Partner	Spouse’s depression	−0.44 (−0.70,−0.18)	0.002	0.004	0.012

Note. *p*-values in the leftmost column are unadjusted for false discovery rate (FDR). Values in the middle column are adjusted within each model. Values in the rightmost column are adjusted across all models. Beta effects are scaled. CI = Confidence interval.

**Table 2 behavsci-14-01161-t002:** Actor–partner Interdependence models with anxiety as the dependent variable.

Independent Variable	APIM Effect	Dependent Variable	B (95% CI)	P_Unadjusted_	P_Adjusted Across Model_	P_Adjusted Across All_
**Neuroticism**						
Patient’s neuroticism	Actor	Patient’s anxiety	0.70 (0.48,0.93)	<0.001	<0.001	<0.001
Spouse’s neuroticism	Actor	Spouse’s anxiety	0.39 (0.15,0.62)	0.002	0.006	0.019
Spouse’s neuroticism	Partner	Patient’s anxiety	−0.18 (−0.42,0.05)	0.14	0.18	0.38
Patient’s neuroticism	Partner	Spouse’s anxiety	0.27 (0.05,0.50)	0.021	0.034	0.10
**Extraversion**						
Patient’s extraversion	Actor	Patient’s anxiety	0.07 (−0.26,0.39)	0.70	0.98	0.87
Spouse’s extraversion	Actor	Spouse’s anxiety	0.04 (−0.26,0.33)	0.81	0.98	0.89
Spouse’s extraversion	Partner	Patient’s anxiety	0.12 (−0.17,0.41)	0.42	0.98	0.68
Patient’s extraversion	Partner	Spouse’s anxiety	0.12 (−0.21,0.44)	0.48	0.98	0.69
**Agreeableness**						
Patient’s agreeableness	Actor	Patient’s anxiety	−0.15 (−0.42,0.12)	0.28	0.46	0.51
Spouse’s agreeableness	Actor	Spouse’s anxiety	−0.22 (−0.53,0.09)	0.17	0.46	0.40
Spouse’s agreeableness	Partner	Patient’s anxiety	−0.22 (−0.53,0.09)	0.18	0.46	0.40
Patient’s agreeableness	Partner	Spouse’s anxiety	−0.08 (−0.34,0.19)	0.59	0.73	0.78
**Conscientiousness**						
Patient’s conscientiousness	Actor	Patient’s anxiety	−0.21 (−0.49,0.06)	0.14	0.69	0.38
Spouse’s conscientiousness	Actor	Spouse’s anxiety	0.01 (−0.29,0.31)	0.96	0.96	0.96
Spouse’s conscientiousness	Partner	Patient’s anxiety	−0.02 (−0.32,0.28)	0.90	0.96	0.94
Patient’s conscientiousness	Partner	Spouse’s anxiety	−0.10 (−0.38,0.18)	0.49	0.96	0.69
**Openness to experiences**						
Patient’s openness to experiences	Actor	Patient’s anxiety	−0.37 (−0.67,−0.07)	0.021	0.10	0.10
Spouse’s openness to experiences	Actor	Spouse’s anxiety	−0.05 (−0.37,0.26)	0.74	0.75	0.87
Spouse’s openness to experiences	Partner	Patient’s anxiety	0.24 (−0.07,0.56)	0.14	0.36	0.38
Patient’s openness to experiences	Partner	Spouse’s anxiety	−0.13 (−0.43,0.18)	0.43	0.71	0.68
**Psychological flexibility**						
Patient’s psychological flexibility	Actor	Patient’s anxiety	−0.56 (−0.85,−0.28)	<0.001	0.001	0.003
Spouse’s psychological flexibility	Actor	Spouse’s anxiety	−0.28 (−0.60,0.03)	0.08	0.21	0.33
Spouse’s psychological flexibility	Partner	Patient’s anxiety	0.05 (−0.26,0.36)	0.76	0.93	0.87
Patient’s psychological flexibility	Partner	Spouse’s anxiety	−0.18 (−0.46,0.11)	0.23	0.38	0.46

Note. *p*-values in the leftmost column are unadjusted for false discovery rate (FDR). Values in the middle column are adjusted within each model. Values in the rightmost column are adjusted across all models. Beta effects are scaled. CI = Confidence interval.

## Data Availability

The data presented in this study are available on request from the corresponding author.

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
