# Peer review of "Actor–Partner Effects of Personality Traits and Psychological Flexibility on Psychological Distress Among Couples Coping with Cancer"

_behavsci, 2024, doi:10.3390/bs14121161_

Round 1
Reviewer 1 Report
Comments and Suggestions for Authors
I believe that it is very valuable to analyze the differences between patients and their spouses in how they cope when diagnosed with cancer. However, I think that the validity of this study needs to be clearly demonstrated in terms of the research design and sample size, etc. If these points are supplemented, I think there will be no major problem in publishing in an academic journal. To summarize these points, they are as follows.
1. First, the rationale for choosing psychological flexibility as a research variable seems to have been adequately presented. However, the validity of choosing other variables, especially personality, needs to be presented more convincingly.
2. Please provide a basis for drawing statistical conclusions from a sample of 46 couples and 92 people.
3. In addition to the design for statistical analysis, you should also present statistical analysis techniques. Provide criteria for drawing statistical conclusions.
4. Please do not present tables as image files, but present tables created directly in a word processing program. It's a good job of visually demonstrating the Actor-Partner effect.
5. The clinical implications of what you found in this study need to be more specific.
Author Response
|
Summary |
|
|
|
Thank you very much for taking the time to review this manuscript. Please find the detailed responses below and the corresponding revisions in track changes in the revised manuscript. |
||
|
|
||
|
Point-by-point response to Comments and Suggestions for Authors
|
||
|
Comments 1: First, the rationale for choosing psychological flexibility as a research variable seems to have been adequately presented. However, the validity of choosing other variables, especially personality, needs to be presented more convincingly. |
||
|
Response 1: Thank you for pointing this out. We have added a reference in the section " Psychological outcomes among couples coping with cancer" which reinforces the importance of expanding knowledge about personality characteristics among a population diagnosed with a disease, rows 60-61:
"Personality refers to emotional and behavioral tendencies that may affect how individuals cope with health challenges".
And in section "Personality traits, psychological flexibility, and psychological distress", rows 84-88:
"Personality is associated with physical health and health behaviors. For example, higher conscientiousness and lower neuroticism are associated with better physical health (measured by disease severity) and health behaviors (i.e. more exercise, healthier diet, less substance use) among general population and cancer patients"
Rochefort, C., Hoerger, M., Turiano, N. A., & Duberstein, P. (2019). Big Five personality and health in adults with and without cancer. Journal of health psychology, 24(11), 1494-1504.
|
||
|
Comments 2: Please provide a basis for drawing statistical conclusions from a sample of 46 couples and 92 people. |
||
|
Response 2: We thank the reviewer for this comment. We have addressed both the statistical validity and the external validity (i.e., the ability to apply conclusions from sample to population) in the limitations section in the discussion, rows 340-346:
“Our sample included 92 participants, which was sufficient in terms of statistical powers as evident by the significant effects achieved in our analyses, even following p-value adjustments for multiple comparisons. However, these results should also be viewed in terms of external validity, i.e., our ability to generalize conclusions to a broader population. The application of our results to couples struggling with cancer should be interpreted with caution, due to the relatively small sample size.”
Comments 3: In addition to the design for statistical analysis, you should also present statistical analysis techniques. Provide criteria for drawing statistical conclusions. Response 3: As a response to the comment, we added the following in the "Statistical analysis" section, rows 185-205 :
"We used the APIM to test actor and partner effects among patients and spouses. In accordance with recommendations regarding APIM best practice [25]. APIM can be modelled via multiple statistical techniques, most recommended are Structural Equation Modelling (SEM), and Multilevel Modelling, also known as Mixed-Effects Modelling. Mixed-effects models account for both fixed effects (which apply to the entire population) and random effects (which account for variability within groups or individuals) in hierarchical or clustered data, such as the couples in our sample [26]. we constructed our model as a multilevel model where fixed effects were the interaction between participant’s role (patient or spouse) and the variables of interest (Big Five personality traits and psychological flexibility), and the random effect was couple’s number. Each variable of interest produced four effects: patient’s actor effect (patients on themselves), patient’s partner effect (spouses on patients), spouse’s actor effect (spouses on themselves), and spouse’s partner effect (patients on spouses). We conducted this analysis using the lmerTest package in R [27]. As some of our variables of interest were intercorrelated, we performed separate models to overcome multicollinearity, such as that in each model a different Big-5 factor was introduced as an independent fixed-effect, and either anxiety or depression was introduced as the fixed effect. A second justification for performing separate models was that introducing all factors into one model would lead to a non-optimal observations-to-variables ratio. We controlled for multiple comparisons with the false discovery rate (FDR), both within each model and across all models [28], using the stats package in R [29]. Results were considered significant for α<.05, after FDR adjustment."
Comments 4: Please do not present tables as image files, but present tables created directly in a word processing program. It's a good job of visually demonstrating the Actor-Partner effect. Response 4: We apologize for this. We have now included the tables in a word format.
Comments 5: The clinical implications of what you found in this study need to be more specific. Response 5: Please see the Summary and conclusions of the study section which reviews the importance of the research findings for psychological screening and suggests interventions in the field of psycho-oncology.
|
||
Reviewer 2 Report
Comments and Suggestions for Authors
My concerns:
The paper is somewhat atheoretical. The authors are identifying a new set of variables (dispositional personality characteristics, psychological flexibility) that are novel as predictors of cancer-related well-being in this literature. However, it remains unclear how these additional variables expand our theoretical understanding of processes related to social support, coping with cancer, and intimate relationships in this popultion. I think the theoretical contribution of the paper needs to be sharpened, both in the introduction and the discussion sections.
There is conceptual overlap between psychological flexibility and openness to experience that the authors never really address. What’s the difference here? Are you just measuring two very similar constructs?
Unfortunately, the low sample size does not really allow for that kind of analysis, but I wonder about statistical higher-order interactions between predictors in the dataset. For example, there are a lack of effects of spouse variables on patient’s psychological distress measures. However, it is possible that factors, such as psychological flexibility moderate associations here. E.g., a spouse’s neuroticism may lead to increased anxiety or depression in the patient only if the patient is low in psychological flexibility. May be some food for thought for the discussion section.
I had a little bit of trouble to identify in Table 2 what the actor and partner effects signified. For example, what does “patient: neuroticism (partner)” mean? Effects of partner’s neuroticism on patient depression? Effects of patient’s neuroticism on partner depression? I think I originally misread this table, which made it much harder to integrate the Table with the Figure and the stats reported in-text. Also, you report stats twice (in tables and in-text) which seems a little bit of an overkill?
In terms of limitations, sample size and power is definitely an issue. I think this needs to be discussed more in the paper.
Comments on the Quality of English LanguageThe paper is overal well-written with some minor issues, but nothing a thorough proof-read cannot take care of. Well done.
Author Response
|
Summary |
|
|
Thank you very much for taking the time to review this manuscript. Please find the detailed responses below and the corresponding revisions in track changes in the revised manuscript. |
|
|
Point-by-point response to Comments and Suggestions for Authors |
|
|
Comments 1 : The paper is somewhat atheoretical. The authors are identifying a new set of variables (dispositional personality characteristics, psychological flexibility) that are novel as predictors of cancer-related well-being in this literature. However, it remains unclear how these additional variables expand our theoretical understanding of processes related to social support, coping with cancer, and intimate relationships in this population. I think the theoretical contribution of the paper needs to be sharpened, both in the introduction and the discussion sections.
|
|
|
Response 1: Thank you for pointing this out. According to the comment that was also made by Reviewer #1 comment #1 , we added a reference in "Personality traits, psychological flexibility, and psychological distress" section which strengthens the importance of expanding knowledge about personality characteristics among a population diagnosed with a disease, rows 84-88 :
"Personality is associated with physical health and health behaviors. For example, higher conscientiousness and lower neuroticism are associated with better physical health (measured by disease severity) and health behaviors (i.e. more exercise, healthier diet, less substance use) among the general population and among cancer patients"
Rochefort, C., Hoerger, M., Turiano, N. A., & Duberstein, P. (2019). Big Five personality and health in adults with and without cancer. Journal of health psychology, 24(11), 1494-1504.
Also, we added a reference in "Summary and conclusions of the study" which Indicates how research findings reinforce and support existing theories, rows 366-370 :
"Our findings align with models emphasizing the importance of personality traits in understanding psychological outcomes in medical contexts. For instance, the 'Interactional stress moderation models'' highlights the influence of personality on both exposure to stressful life circumstances and the availability of coping resources."
Smith, T. W. (2006). Personality as risk and resilience in physical health. Current directions in psychological science, 15(5), 227-231.
|
|
|
Comments 2: There is conceptual overlap between psychological flexibility and openness to experience that the authors never really address. What’s the difference here? Are you just measuring two very similar constructs?
|
|
|
Response 2: We agree that these constructs share similarity. However, they do differ with relation to them being more balanced or to step out of the comfort zone. We have added a reference in "Personality traits, psychological flexibility, and psychological distress" section to emphasize this point, rows 121-127:
"Openness to experience, one of the Big Five personality traits, involves some aspect of cognitive flexibility like the willingness to engage with new ideas. However, the main characteristics of this trait is one having an imagination, curiosity, and a preference for variety. Psychological flexibility, on the other hand, refers to the ability to adapt to situational demands and maintain balance in the face of challenges via the acceptance of thoughts and feelings without judgment and aligning actions with personal values."
Bond, F. W., Hayes, S. C., Baer, R. A., Carpenter, K. M., Guenole, N., Orcutt, H. K., ... & Zettle, R. D. (2011). Preliminary psychometric properties of the Acceptance and Action Questionnaire–II: A revised measure of psychological inflexibility and experiential avoidance. Behavior therapy, 42(4), 676-688.
P. T. Costa and R. R. McCrae, “Normal personality assessment in clinical practice: The NEO Personality Inventory.,” Psychol Assess, vol. 4, no. 1, pp. 5–13, Mar. 1992, doi: 10.1037/1040-3590.4.1.5.
Comments 3: Unfortunately, the low sample size does not really allow for that kind of analysis, but I wonder about statistical higher-order interactions between predictors in the dataset. For example, there are a lack of effects of spouse variables on patient’s psychological distress measures. However, it is possible that factors, such as psychological flexibility moderate associations here. E.g., a spouse’s neuroticism may lead to increased anxiety or depression in the patient only if the patient is low in psychological flexibility. May be some food for thought for the discussion section.
Response 3: We agree with the reviewer and added a limitation in the discussion, row 346-352:
“Another implication of our small sample size is that higher-order interactions, such as the potential moderation of actor and partner’s psychological flexibility on the association between neuroticism and mental health outcomes, could not be investigated with sufficient statistical power. Therefore, it is reasonable to suggest that in a larger sample, and in an analysis in which the diversity of the groups is taken into consideration, it would be possible to obtain additional significant results and examine complex relationships between multiple variables and their dynamics.”
Comments 4: I had a little bit of trouble to identify in Table 2 what the actor and partner effects signified. For example, what does “patient: neuroticism (partner)” mean? Effects of partner’s neuroticism on patient depression? Effects of patient’s neuroticism on partner depression? I think I originally misread this table, which made it much harder to integrate the Table with the Figure and the stats reported in-text. Also, you report stats twice (in tables and in-text) which seems a little bit of an overkill?
Response 4: We thank the reviewer for this important comment. We have edited the tables and improved their clarity.
Comments 5: In terms of limitations, sample size and power is definitely an issue. I think this needs to be discussed more in the paper. Response 2: Thanks for this comment. We expanded the reference to this topic in the limitations section.
|
|
|
4. Response to Comments on the Quality of English Language |
|
|
Point 1: The paper is overal well-written with some minor issues, but nothing a thorough proof-read cannot take care of. Well done. |
|
|
Response 1: We re-edited the paper.
|
|